# *Bcl2l1* Deficiency in Osteoblasts Reduces the Trabecular Bone Due to Enhanced Osteoclastogenesis Likely through Osteoblast Apoptosis

**DOI:** 10.3390/ijms242417319

**Published:** 2023-12-10

**Authors:** Takeshi Moriishi, Yosuke Kawai, Ryo Fukuyama, Yuki Matsuo, You-Wen He, Haruhiko Akiyama, Izumi Asahina, Toshihisa Komori

**Affiliations:** 1Department of Cell Biology, Nagasaki University Graduate School of Biomedical Sciences, Nagasaki 852-8588, Japan; moriishi@nagasaki-u.ac.jp (T.M.); ysmatsuo@nagasaki-u.ac.jp (Y.M.); 2Department of Regenerative Oral Surgery, Medical and Dental Sciences, Nagasaki University Graduate School of Biomedical Sciences, Nagasaki 852-8588, Japan; kawaiyosuke33@yahoo.co.jp; 3Laboratory of Pharmacology, Hiroshima International University, Kure 737-0112, Japan; r-fukuya@hirokoku-u.ac.jp; 4Department of Molecular Bone Biology, Nagasaki University Graduate School of Biomedical Sciences, Nagasaki 852-8588, Japan; 5Department of Immunology, Duke University Medical Center, Durham, NC 27710, USA; youwen.he@duke.edu; 6Department of Orthopedic Surgery, Graduate School of Medicine, Gifu University, Gifu 501-1194, Japan; hakiyama@gifu-u.ac.jp; 7Department of Oral and Maxillofacial Surgery, Juntendo University School of Medicine, 2-1-1 Hongo, Bunkyo-ku, Tokyo 113-8431, Japan; i.asahina.pn@juntendo.ac.jp

**Keywords:** Bcl2l1, Bcl-XL, apoptosis, ATP, Tnfsf11, osteoblast, osteoclast, bone resorption

## Abstract

Bcl2l1 (Bcl-XL) belongs to the Bcl-2 family, Bcl2 and Bcl2-XL are major anti-apoptotic proteins, and the apoptosis of osteoblasts is a key event for bone homeostasis. As the functions of Bcl2l1 in osteoblasts and bone homeostasis remain unclear, we generated osteoblast-specific *Bcl2l1*-deficient (*Bcl2l1*^fl/flCre^) mice using 2.3-kb *Col1a1* Cre. Trabecular bone volume and the trabecular number were lower in *Bcl2l1*^fl/flCre^ mice of both sexes than in *Bcl2l1*^fl/fl^ mice. In bone histomorphometric analysis, osteoclast parameters were increased in *Bcl2l1*^fl/flCre^ mice, whereas osteoblast parameters and the bone formation rate were similar to those in *Bcl2l1*^fl/fl^ mice. TUNEL-positive osteoblastic cells and serum TRAP5b levels were increased in *Bcl2l1*^fl/flCre^ mice. The deletion of *Bcl2l1* in osteoblasts induced *Tnfsf11* expression, whereas the overexpression of *Bcl-XL* had no effect. In a co-culture of *Bcl2l1*-deficient primary osteoblasts and wild-type bone-marrow-derived monocyte/macrophage lineage cells, the numbers of multinucleated TRAP-positive cells and resorption pits increased. Furthermore, serum deprivation or the deletion of *Bcl2l1* in primary osteoblasts increased apoptosis and ATP levels in the medium. Therefore, the reduction in trabecular bone in *Bcl2l1*^fl/flCre^ mice may be due to enhanced bone resorption through osteoblast apoptosis and the release of ATP from apoptotic osteoblasts, and *Bcl2l1* may inhibit bone resorption by preventing osteoblast apoptosis.

## 1. Introduction

Osteoblast apoptosis plays an important role in bone development and maintenance. Between 60 and 80% of osteoblasts that originally assemble at the resorption pit are estimated to die by apoptosis. Furthermore, bone loss caused by sex steroid deficiency, glucocorticoid excess, or aging is caused in part by osteoblast apoptosis, and parathyroid hormone (PTH), bisphosphonate, and calcitonin exert anabolic effects on bone by inhibiting osteoblast apoptosis [1,2,3,4,5,6,7,8]. The intrinsic pathway, also called the BCL-2-regulated or mitochondrial pathway, is activated by various developmental cues or cytotoxic insults, such as viral infection, DNA damage, and growth-factor deprivation. This pathway is strictly controlled by the BCL-2 family of proteins and predominantly leads to the activation of caspase-9 [9]. Bcl2 subfamily proteins, including Bcl2 and Bcl-XL, inhibit apoptosis by preventing the release of caspase activators from mitochondria through the inhibition of Bax subfamily proteins [10].

A previous study reported the enhancement of osteoclast apoptosis and an increase in bone mass in *Bcl2*-deficient (*Bcl2*^−/−^) mice, indicating that Bcl2 attenuated osteoclast apoptosis [11,12]. However, the function of Bcl2 in osteoblasts is controversial. *Bcl2* deficiency did not affect osteoblasts [11], impair osteoblast function [12], or enhance osteoblast differentiation [13]. This discrepancy may be attributed to enhanced apoptosis in *Bcl2*^−/−^ osteoblasts during their cultivation, which reduced cell density and decelerated osteoblast differentiation in vitro [13,14]. 

Five isoforms of Bcl2l1 (Bcl-XL, Bcl-XS, Bcl-Xγ, Bcl-Xβ, and Bcl-X△TM) have been identified to date, and Bcl-XL is the major isoform of Bcl2l1 [15,16,17,18,19]. *Bcl2l1*^−/−^ mice die on embryonic day (E) 13.5 and exhibit severe defects in erythropoiesis and neuronal development [20]. The osteoclast-specific deletion of *Bcl2l1* using Cre knock-in mice in the cathepsin K locus did not affect osteoclast numbers, but increased bone resorption and resulted in osteopenia. Bcl2l1 in osteoclasts is suggested to attenuate osteoclastic bone-resorbing activity through a decrease in the production of extracellular matrix proteins, such as vitronectin and fibronectin [21]. Therefore, Bcl2 and Bcl2l1 have distinct functions in osteoclasts. 

We recently showed that *Bcl2* deficiency up-regulated p53 in osteoblasts, which induced Pten and Igfbp3, inactivated Akt, leading to the activation of FoxO1 and FoxO3a, and accelerated osteoblast differentiation [13]. The overexpression of *Bcl2* in osteoblasts impaired osteocyte process formation, leading to a reduction in the number of canaliculi and osteocyte apoptosis. In contrast, the overexpression of *Bcl-XL* in osteoblasts increased the volumes of both the trabecular and cortical bone with normal structures and these increases were maintained during aging. Therefore, the functions of Bcl2 and Bcl-XL in osteoblasts may also differ. 

Apoptotic cells have been shown to release adenosine triphosphate (ATP) through pannexin 1 (Panx1) as a ‘find me’ signal in the earliest stage of death to recruit phagocytes [22]. ATP binds to P2Y1 receptors on osteoblasts and enhances *Tnfsf11* expression [23]. ATP also activates NF-κB signaling in osteoclasts through P2X7 receptors on osteoclasts, and ATP drives osteoclast fusion through P2X7 receptors [24,25]. Therefore, the activation of P2Y and P2X receptors by ATP leads to enhanced bone resorption [26]. 

We herein demonstrated that osteoblast numbers and bone formation were maintained in osteoblast-specific *Bcl2l1*-deficient mice, whereas osteoclastogenesis was accelerated through the induction of *Tnfsf11* expression in osteoblasts likely through the release of ATP from apoptotic osteoblasts, resulting in a reduction in trabecular bone. 

## 2. Results

### 2.1. Generation of Osteoblast-Specific Bcl2l1 Conditional Knock-Out Mice

To examine the expression levels of *Bcl2l1* in various tissues, *Bcl2l1* expression was examined using RNA from bone marrow, femur, brain, lung, heart, liver, stomach, kidney, and spleen of 10-week-old mice. *Bcl2l1* expression in the femur was moderate among these tissues (Figure 1A). To specifically delete *Bcl2l1* in osteoblasts, we confirmed Cre expression in 2.3-kb *Col1a1* promoter Cre transgenic mice. The 2.3-kb *Col1a1* promoter Cre transgenic mice were mated with β-galactosidase reporter mice under the control of the CAG promoter. In double transgenic mice, β-galactosidase activity was specifically detected in osteoblasts (Figure 1B,C). *Bcl2l1*^fl/+^ mice were crossed with Cre transgenic mice and *Bcl2l1*^fl/flCre^ mice were generated. Lower Bcl2l1 protein levels in *Bcl2l1*^fl/flCre^ femurs than in *Bcl2l1*^fl/fl^ femurs were confirmed by Western blotting (Figure 1D,E). Body weights in *Bcl2l1*^fl/flCre^ mice were similar to those in *Bcl2l1*^fl/fl^ mice of both sexes (Figure 1F).

### 2.2. Reduction in Bone Mass in Bcl2l1^fl/flCre^ Mice

In the μCT analysis, the trabecular bone volume and the trabecular number in the femur were lower in *Bcl2l1*^fl/flCre^ mice than in *Bcl2l1*^fl/fl^ mice of both sexes (Figure 2A,B,E). Trabecular thickness and trabecular bone mineral density (BMD) were lower in male *Bcl2l1*^fl/flCre^ mice than in male *Bcl2l1*^fl/fl^ mice (Figure 2E). At the mid-diaphysis of the femoral cortical bone, cortical thickness, endosteal perimeters, and cortical BMD in *Bcl2l1*^fl/flCre^ mice were similar to those in *Bcl2l1*^fl/fl^ mice in both sexes, while the periosteal perimeter was shorter in *Bcl2l1*^fl/flCre^ mice than in *Bcl2l1*^fl/fl^ mice among males, but not females (Figure 2C,D,F).

### 2.3. Increases in Osteoclast Parameters in Bcl2l1^fl/flCre^ Mice in the Bone Histomorphometric Analysis

The bone histomorphometric analysis was performed using male femurs at 10 weeks of age. In the trabecular bone, osteoblast parameters, including the osteoid surface, osteoid thickness, osteoblast surface, and osteoblast number, and the parameters for bone formation, including the mineral apposition rate, mineralizing surface, and bone formation rate, were similar between *Bcl2l1*^fl/flCre^ and *Bcl2l1*^fl/fl^ mice, whereas the osteoclast parameters, including the osteoclast surface and eroded surface, were increased in *Bcl2l1*^fl/flCre^ mice (Figure 3A). In the cortical bone, the mineralizing surface in the periosteum, but not in the endosteum, was slightly larger in *Bcl2l1*^fl/flCre^ mice than in *Bcl2l1*^fl/fl^ mice, while the mineral apposition rate and bone formation rate in both the periosteum and endosteum were similar in *Bcl2l1*^fl/flCre^ mice and *Bcl2l1*^fl/fl^ mice (Figure 3B,C).

### 2.4. Increased Osteoblast Apoptosis and Serum TRAP5b Levels in Bcl2l1^fl/flCre^ Mice

Since the number of osteoblasts was not decreased in *Bcl2l1*^fl/flCre^ mice, the proliferation of osteoblasts was examined by BrdU labeling. The frequencies of BrdU-positive cells were similar between *Bcl2l1*^fl/flCre^ and *Bcl2l1*^fl/fl^ mice (Figure 4A,B). We then examined osteoblast apoptosis by TUNEL staining. The frequency of TUNEL-positive osteoblastic cells was higher in *Bcl2l1*^fl/flCre^ mice than in *Bcl2l1*^fl/fl^ mice (Figure 4C,D). The level of the serum marker for bone formation, osteocalcin, was similar in *Bcl2l1*^fl/flCre^ mice and *Bcl2l1*^fl/fl^ mice, whereas that of the serum marker for bone resorption, tartrate-resistant acid phosphatase 5b (TRAP5b), was higher in *Bcl2l1*^fl/flCre^ mice than in *Bcl2l1*^fl/fl^ mice (Figure 4E,F).

### 2.5. Osteoblastogenesis and Osteoclastogenesis In Vitro

Primary osteoblasts from *Bcl2l1*^fl/fl^ mice were infected with the green fluorescent protein (GFP)- or Cre-expressing adenovirus. Osteoblast differentiation was evaluated by alkaline phosphatase (ALP) and von Kossa staining. Staining was similar between cells infected with the GFP- and Cre-expressing adenoviruses, indicating that the deletion of *Bcl2l1* did not affect osteoblast differentiation in the early or late stage (Figure 5A–D). Since osteoclast parameters and serum TRAP5b levels were increased in *Bcl2l1*^fl/flCre^ mice, *Tnfsf11* and *Tnfrsf11b* expression levels were examined by real-time RT-PCR. The expression levels of *Tnfsf11*, but not *Tnfrsf11b*, were increased and the *Tnfsf11*/*Tnfrsf11b* ratio was higher with the Cre-expressing adenovirus than with the GFP-expressing adenovirus (Figure 5E). The expression of *Tnfsf11* and *Tnfrsf11b* was also examined in *Bcl-XL* transgenic (tg) mice under the control of the 2.3-kb *Col1a1* promoter [27] using the newborn calvaria and osteoblast fraction from femurs and tibiae at 6 weeks of age. Although *Tnfrsf11b* expression levels in the calvaria were slightly higher in *Bcl-XL* tg mice than in wild-type mice, *Tnfsf11* expression levels and the *Tnfsf11*/*Tnfrsf11b* ratio were similar between wild-type and *Bcl-XL* tg mice in both the newborn calvaria and osteoblast fraction from femurs and tibiae (Figure 5F,G). Similar results were also obtained for primary osteoblasts from *Bcl-XL* tg mice (Figure 5H).

*Bcl2l1*^fl/fl^ primary osteoblasts, which were infected with the GFP- or Cre-expressing adenovirus, were co-cultured with bone-marrow-derived monocyte/macrophage lineage cells (BMMs) from *Bcl2l1*^fl/fl^ mice. Multinucleated TRAP-positive cells were higher in the co-culture using primary osteoblasts infected with the Cre-expressing adenovirus than in that with the GFP-expressing adenovirus (Figure 6A,B). The pit assay was performed on the co-culture of primary osteoblasts from *Bcl2l1*^fl/fl^ or *Bcl2l1*^fl/flCre^ mice with BMMs from *Bcl2l1*^fl/fl^ mice. The area of resorption pits in the co-culture using *Bcl2l1*^fl/flCre^ primary osteoblasts was larger than in that using *Bcl2l1*^fl/fl^ primary osteoblasts (Figure 6C,D).

### 2.6. Increases in Apoptosis and ATP Release by the Bcl2l1 Deletion

To confirm that ATP was released in the medium by osteoblast apoptosis, primary osteoblasts from wild-type mice were cultured in the presence or absence of serum for 24 h. In the absence of serum, TUNEL-positive cells were increased and ATP release in the medium, which was measured by luciferase, was also enhanced (Figure 7A–C). Elevations were observed in *Tnfsf11* expression levels and the *Tnfsf11/Tnfrsf11b* ratio in the absence of serum (Figure 7D). Moreover, *Tnfsf11* expression levels and the *Tnfsf11/Tnfrsf11b* ratio were increased after the culture for 24 h in serum-free conditioned medium supplemented with 10% FBS, indicating that released factors were responsible for the induction of *Tnfsf11* expression (Figure 7E). 

The number of TUNEL-positive cells in *Bcl2l1*^fl/fl^ primary osteoblasts infected with the Cre-expressing adenovirus was higher than in those infected with the GFP-expressing adenovirus (Figure 7F,G). ATP levels in the medium of *Bcl2l1*^fl/fl^ primary osteoblasts infected with the Cre-expressing adenovirus were higher than in that of those infected with the GFP-expressing adenovirus, and this increase was eliminated by the addition of apyrase, which catalyzes the hydrolysis of ATP (Figure 7H).

Trabecular bone volume was reduced in *Bcl2l1*^fl/flCre^ mice due to increased bone resorption. The deletion of *Bcl2l1* in osteoblasts did not affect osteoblast differentiation, but enhanced osteoclastogenesis through the induction of *Tnfsf11* expression in osteoblasts. However, Bcl2l1 did not appear to exert intrinsic effects on the expression of *Tnfsf11* because the overexpression of *Bcl-XL* in osteoblasts did not affect *Tnfsf11* expression. Osteoblast apoptosis was increased in *Bcl2l1*^fl/flCre^ mice, and the deletion of *Bcl2l1* in vitro increased osteoblast apoptosis and ATP, which is released from apoptotic osteoblasts and induces *Tnfsf11* expression and osteoclastogenesis [28], in the medium. These results suggest that the *Bcl2l1* deletion in osteoblasts increased apoptosis, apoptotic osteoblasts released ATP, and ATP enhanced osteoclastogenesis, which promoted bone resorption.

Although osteoblast apoptosis was increased in *Bcl2l1*^fl/flCre^ mice, osteoblast parameters and the bone formation rate in the bone histomorphometric analysis and serum osteocalcin levels were not reduced. In contrast, osteoclast parameters in the bone histomorphometric analysis and serum TRAP5b levels were increased in *Bcl2l1*^fl/flCre^ mice compared with *Bcl2l1*^fl/fl^ mice. Therefore, the reduction in trabecular bone volume in *Bcl2l1*^fl/flCre^ mice was due to enhanced bone resorption. The deletion of *Bcl2l1* in primary osteoblasts increased the expression of *Tnfsf11*, whereas the overexpression of *Bcl-XL* had no effect. Moreover, bone resorption in osteoblast-specific *Bcl-XL* tg mice was similar to that in wild-type mice [27]. Therefore, increased bone resorption in *Bcl2l1*^fl/flCre^ mice and enhanced osteoclastogenesis and *Tnfsf11* induction by the *Bcl2l1* deletion in osteoblasts were not considered to have been caused by the intrinsic effects of Bcl2l1. Since the *Bcl2l1* deletion in osteoblasts increased apoptosis and ATP release into the medium, the deletion of *Bcl2l1* was considered to have enhanced osteoclastogenesis through the release of ATP from apoptotic osteoblasts. Apoptotic osteoblasts are quickly removed by phagocytes; therefore, it is difficult to observe the induction of Tnfsf11 in neighboring osteoblasts in vivo. In the case of apoptotic osteocytes, which are not phagocytosed, the induction of Tnfsf11 was observed in neighboring osteocytes and osteoblasts [29,30]. Therefore, ATP released from apoptotic osteoblasts may have induced Tnfsf11 expression by binding to P2Y receptors on neighboring osteoblasts in vivo [23]. ATP will also bind to P2X receptors on neighboring osteoclast progenitors and osteoclasts and enhance their differentiation and activity [24,25,31].

Since the increase in osteoblast apoptosis did not affect osteoblast parameters or the bone formation rate in the bone morphometric analysis or the serum level of osteocalcin in *Bcl2l1*^fl/flCre^ mice, some mechanisms may have compensated for osteoblast death. P2Y receptors has been shown to positively or negatively function in the proliferation, differentiation, and functions of osteoblasts [32]. A previous study demonstrated that ATP potentiated parathyroid hormone-mediated signaling though P2Y1 receptors, ATP released by low-intensity ultrasound induced osteoblastogenesis through P2Y1 receptors, and an antagonist of P2Y12 receptors, which was selective for ADP and ATP, inhibited osteoblast proliferation and differentiation and reduced trabecular bone volume [33,34,35]. Therefore, ATP released from apoptotic osteoblasts may have enhanced osteoblastogenesis and compensated for the loss of osteoblasts.

In conclusion, *Bcl2l1* was involved in the survival of osteoblasts, the *Bcl2l1* deletion in osteoblasts appeared to enhance bone resorption by inducing osteoblast apoptosis, and ATP released from apoptotic osteoblasts may be responsible for enhanced bone resorption. This is the first study to suggest that osteoblast apoptosis reduces bone mass by enhancing bone resorption, although the mechanism shown here by in vitro studies needs to be confirmed in vivo. Since the overexpression of *Bcl-XL* in osteoblasts has been shown to enhance bone formation and increase the trabecular and cortical bone by preventing osteoblast apoptosis [27], the inhibition of osteoblast apoptosis is important not only for enhancing bone formation, but also for suppressing bone resorption.

## 4. Materials and Methods

### 4.1. Mice

Cre transgenic mice driven by the 2.3-kb *Col1a1* promoter, CAG-CAT-LacZ transgenic mice [36], and *Bcl2l1* flox mice [37] were kindly provided by Dr. H. Akiyama (Gifu University, Gifu, Japan), Dr. J. Miyazaki (Osaka University, Suita, Japan), and Dr. Y. He (Duke University, Durham, NC, USA), respectively. *Bcl-XL* tg mice under the control of the 2.3-kb *Col1a1* promoter were generated as previously described [27]. Before the initiation of the present study, all experimental protocols were reviewed and approved by the Animal Care and Use Committee of Nagasaki University Graduate School of Biomedical Sciences (No. 1903131520-9). Animals were housed three per cage in a pathogen-free environment on a 12 h light cycle at 22 ± 2 °C, with standard chow (CLEA Japan, Tokyo, Japan) and free access to tap water.

### 4.2. Real-Time RT-PCR and Western Blot Analyses

Total RNA was extracted from bone marrow, femur, brain, lung, heart, liver, stomach, kidney, and spleen of wild-type mice at 10 weeks of age, newborn calvariae of wild-type and *Bcl-XL* tg mice, and from osteoblast-enriched samples from the femurs and tibiae of wild-type and *Bcl-XL* tg mice at 6 weeks of age using ISOGEN (Wako, Osaka, Japan). To obtain the osteoblast-enriched samples, muscle, connective tissue, and periosteum were removed from the femurs and tibiae, and the bones were cut at the metaphyses. After the hematopoietic cells in the diaphyses of femurs and tibiae were flushed out with PBS, the osteoblast-enriched cells were collected using a microintertooth brush (Kobayashi Pharmaceutical Co., Ltd., Osaka, Japan). Total RNA was also extracted from primary osteoblasts using ISOGEN. Real-time PT-PCR was performed using THUNDERBIRD SYBR qPCR Mix (Toyobo, Osaka, Japan) and Light Cycler 480 real-time PCR system (Roche Diagnostics, Tokyo, Japan). Primer sequences were as follows: *Actb*, 5′-CCACCCGCGAGCACAGCTTC-3′ and 5′-TTGTCGACGACCAGCGCAGC-3′, *Bcl2l1*, 5′-GCAGGTATTGGTGAGTCG-3′ and 5′-GGCTGCTGCATTGTTCCC-3′; *Tnfsf11*, 5′-CAAGCTCCGAGCTGGTGAAG-3′ and 5′-CCTGAACTTTGAAAGCCCCA-3′; *Tnfrsf11b*, 5′-AAGAGCAAACCTTCCAGCTGC-3′ and 5′-CACGCTGCTTTCACAGAGGTC-3*′*. We normalized values to those of *Actb*. Protein was extracted from femurs and tibiae, and Western blot analysis was performed using anti-Cre (Covance, Princeton, NJ, USA) and anti-Bcl-XL (Cell Signaling Technology, Danvers, MA, USA) antibodies.

### 4.3. X-Gal Staining

X-gal staining was performed using mice at 4 weeks of age. Mice were sacrificed and the long bones were dissected. They were fixed for 30 min in 0.1 M phosphate buffer with 5 mM EGTA, 2 mM MgCl_2_, and 4% paraformaldehyde, and rinsed three times for 15 min each with 0.1 M phosphate buffer containing 5 mM EGTA, 2 mM MgCl_2_, 0.02% Nonidet P-40, and 0.01% sodium deoxycholate. They were then stained with the same rinsing solution supplemented with 0.5 mg/mL of X-gal, 10 mM potassium ferrocyanide, and 10 mM potassium ferricyanide. After staining, samples were washed in the same rinsing solution for 24 h.

### 4.4. Micro-CT Analysis

A quantitative micro-CT analysis was performed with a micro-CT system (R_mCT; Rigaku Corporation, Tokyo, Japan). Data from scanned slices were used for a three-dimensional analysis to calculate femoral morphometric parameters using the software (Ratoc, TRI-3D BON version R9.02.00.0-H-64, RATOC Systems, Inc., Osaka, Japan). Trabecular bone parameters were measured on a distal femoral metaphysis. Approximately 2.4 mm (0.5 mm from the growth plate) was cranio-caudally scanned, and 200 slices were taken at 12 μm intervals. Cortical thickness was measured at the mid-diaphyses of femurs. We used a threshold value of 275 to binarize the spongiosa and cortex.

### 4.5. Histological Analysis

To assess dynamic histomorphometric indices, mice were injected with calcein 13 d and 3 d before sacrifice. For bone histomorphometric analyses, femurs were fixed with 70% ethanol, and the undecalcified bones were embedded in glycol methacrylate. Three μm longitudinal sections from the distal parts of femurs were stained with toluidine blue and analyzed using a semiautomated system (Bone Histomorphometric System, System Supply, Nagano, Japan). The trabecular bone was measured in the secondary spongiosa commencing 480 μm below the growth plate and extending proximally for 560 μm. Calcein labeling in the cortical bone was measured using 50 μm polishing sections from the mid-diaphyses of femurs by cellSens Standard (OLYMPUS, Tokyo, Japan).

In histological analyses of long bones, mice were sacrificed and fixed in 4% paraformaldehyde/0.01 M phosphate-buffered saline, and the long bones were decalcified in 10% EDTA (pH = 7.4) and embedded in paraffin. Sections (thickness of 3 μm) were stained for TUNEL using the ApopTag^®^ system (Millipore, Darmstadt, Germany). In the BrdU incorporation study, 2-week-old mice were injected intraperitoneally with 100 μg BrdU/g body weight and sacrificed 1 h later. Sections were stained with a BrdU staining kit (Zymed, San Francisco, CA, USA). In the counting of TUNEL-positive or BrdU-positive osteoblastic cells, only cells in the distal primary spongiosa of femurs, which were recognized as osteoblastic cells from their morphology and attachment to trabecular bone, were counted.

### 4.6. Serum Testing

The serum levels of osteocalcin and TRAP5b were measured using a Mouse Osteocalcin Enzyme Immunoassay Kit (BTI, Stoughton, MA, USA) and Mouse TRAP Assay (Immunodiagnostic Systems, Boldon, UK), respectively.

### 4.7. Cell Culture

Primary osteoblasts were isolated from newborn calvariae by sequential digestion with 0.1% collagenase A and 0.2% dispase. Osteoblastic cells from the third to fifth fractions were pooled, plated on 24-well plates at a density of 1 × 10^5^/well in α-MEM supplemented with 10% fetal bovine serum (FBS). At confluency, cells were infected with an adenovirus expressing GFP or Cre at a multiplicity of infection (MOI) of 10 for 3 h. After 24 h, cells were plated on 24-well plates at a density of 1 × 10^5^/well, and after the culture for 24 h, the medium was changed to osteogenic medium containing 50 μg/mL ascorbic acid and 10 mM β-glycerophosphate. ALP and von Kossa staining was performed after the culture for 3 and 12 days, respectively, in the osteogenic medium. Staining for ALP activity and mineralization (von Kossa) was performed as previously described [13]. ALP and von Kossa staining was quantified using ImageJ version 1.47. Total RNA was extracted after the culture for 4 days in the osteogenic medium.

### 4.8. In Vitro Osteoclastogenesis

BMMs were isolated by density gradient centrifugation using Ficoll-Paque^TM^ (GE Healthcare, Tokyo, Japan) from the bone marrow of *Bcl2l1*^fl/fl^ mice at 12 weeks of age. Primary osteoblasts from the newborn calvariae of *Bcl2l1*^fl/fl^ mice were infected with an adenovirus expressing GFP or Cre for 3 h. After 24 h, cells at 1.5 × 10^5^ cells/cm^2^ were co-cultured with BMMs at 2.5 × 10^5^ cells/cm^2^ in α-MEM containing 10% FBS in the presence of 50 μg/mL of ascorbic acid and 10^−8^ M 1α, 25 (OH)_2_D_3_ in 48-well plates for 4 days. Osteoclast formation was evaluated by TRAP staining. Cultured cells were fixed with 4% paraformaldehyde for 30 min, washed in 0.01M PBS at room temperature for 5 min, and incubated in acetate buffer (pH 5.0) containing, N, N-dimethylformamide, naphthol AS-MX phosphate, fast red AL salt, and 50 mM sodium tartrate at 37 °C for 15 min.

To evaluate bone resorption, primary osteoblasts from the newborn calvariae of *Bcl2l1*^fl/fl^ or *Bcl2l1*^fl/flCre^ mice at a density of 5 × 10^5^ cells/cm^2^ were co-cultured with BMMs at 5 × 10^5^ cells/cm^2^ on dentin slices (Wako) in 48-well plates for 8 days. The wells were treated by ultrasonication in 1 M ammonia water, dried, and observed by scanning microscopy (HITACHI, Miniscope TM-1000, Tokyo, Japan).

### 4.9. Measurement of ATP

Primary osteoblasts from *Bcl2l1*^fl/fl^ mice were plated on 48-well plates at a density of 1 × 10^5^/well. At confluency, cells were infected with an adenovirus expressing GFP or Cre at a MOI of 10 for 3 h. The medium was changed to fresh medium containing 1 U/mL of apyrase (NACALAI TESQUE, Inc., Kyoto, Japan). After a culture for 72 h, ATP levels in the supernatant were measured using an Intracellular ATP detection kit (IC100, TOYO B-Net, Co., Ltd. Tokyo, Japan).

### 4.10. Serum-Free Culture

Primary osteoblasts from *Bcl2l1*^fl/fl^ mice were plated on 24-well plates at a density of 1 × 10^5^/well and cultured in α-MEM with or without 10% FBS for 24 h, and TUNEL staining was performed using the ApopTag^®^ system (Millipore) or RNA was extracted from cells. The supernatant was used to measure ATP. Furthermore, the supernatant with 10% FBS was used for the culture of primary osteoblasts, the supernatant without 10% FBS was used as conditioned medium in the culture of primary osteoblasts after the addition of FBS to 10%, and RNA was extracted after the culture for 24 h.

### 4.11. Statistical Analysis

Statistical analyses were performed by the Student’s *t*-test in the comparison of two groups except for Figure 7H, in which the four groups were compared by the Tukey–Kramer test, using BellCurve version 4.05 for Excel (Social Survey Research Information Co., Ltd., Tokyo, Japan). Data are presented as the mean ± S.D. A *p*-value < 0.05 was considered to be significant.

## Figures and Tables

**Figure 1 ijms-24-17319-f001:**
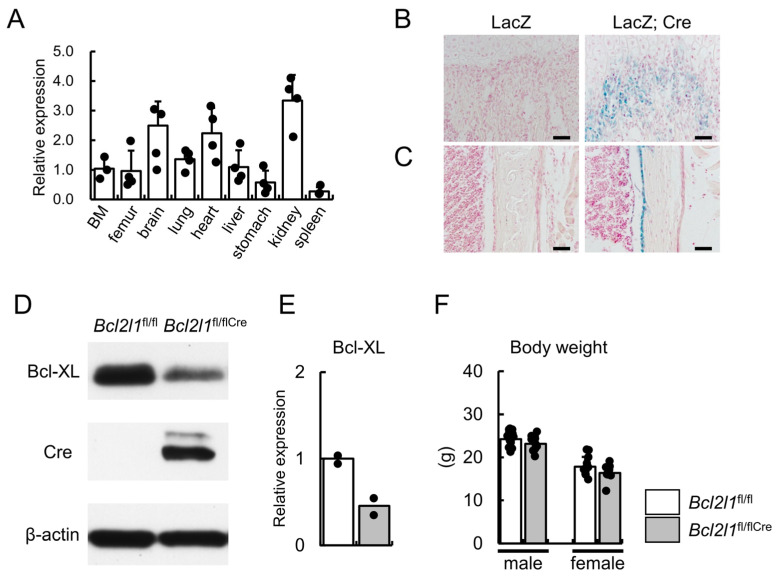
Generation of osteoblast-specific *Bcl2l1* conditional knock-out mice. (**A**) Real-time RT-PCR analysis. *Bcl2l1* expression was examined using RNA from bone marrow, the femur, brain, lung, heart, liver, stomach, kidney, and spleen of wild-type mice at 10 weeks of age. (**B**,**C**) β-Galactosidase staining of sections from the femoral metaphyses (**B**) and femoral diaphyses (**C**) of CAG-LacZ transgenic mice (LacZ) and CAG-LacZ: Cre double transgenic mice (LacZ; Cre) at 4 weeks of age. Scale bars: 50 μm (**B**,**C**). (**D**,**E**) Western blot analysis using anti-Bcl-XL and anti-Cre antibodies. β-actin was used as an internal control. The intensities of the Bcl-XL bands were normalized against β-actin, normalized values in the average of *Bcl2l1*^fl/fl^ mice were set as 1, and relative levels are shown in (**E**). Proteins were extracted from femurs at 5 weeks of age. *n* = 2. (**F**) Body weights of male and female *Bcl2l1*^fl/fl^ and *Bcl2l1*^fl/flCre^ mice at 10 weeks of age. *n* = 14 (male *Bcl2l1*^fl/fl^), *n* = 11 (male *Bcl2l1*^fl/flCre^), *n* = 11 (female *Bcl2l1*^fl/fl^), *n* = 12 (female *Bcl2l1*^fl/flCre^).

**Figure 2 ijms-24-17319-f002:**
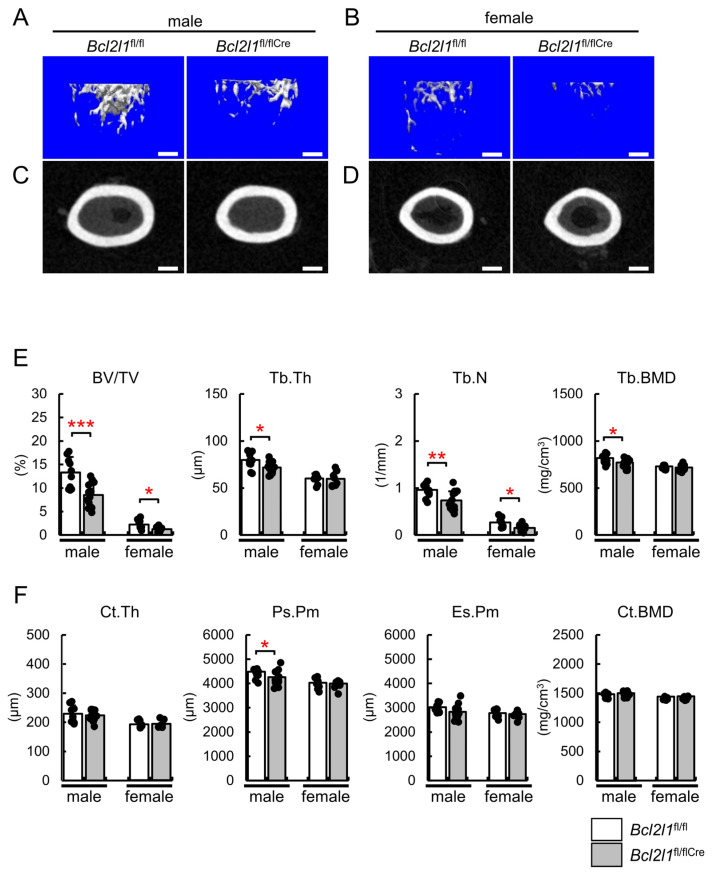
Micro-CT analysis. (**A**–**D**) Three-dimensional images of the distal metaphysis (**A**,**B**) and images of the cortical bone at the mid-diaphysis (**C**,**D**) of femurs from male (**A**,**C**) and female (**B**,**D**) *Bcl2l1*^fl/fl^ and *Bcl2l1*^fl/flCre^ mice at 10 weeks of age. Scale bars: 0.5 mm (**A**,**B**); 0.3 mm (**C**,**D**). (**E**,**F**) Quantification of trabecular bone (**E**) and cortical bone (**F**) parameters in male and female *Bcl2l1*^fl/fl^ and *Bcl2l1*^fl/flCre^ mice at 10 weeks of age. These parameters include trabecular bone volume (bone volume/tissue volume, BV/TV), trabecular thickness (Tb.Th), the trabecular number (Tb.N), trabecular bone mineral density (Tb.BMD), cortical thickness (Ct.Th), the periosteal perimeter (Ps.Pm), endosteal perimeter (Es.Pm), and cortical bone mineral density (Ct.BMD). *n* = 10 (male *Bcl2l1*^fl/fl^), *n* = 15 (male *Bcl2l1*^fl/flCre^), *n* = 8 (female *Bcl2l1*^fl/fl^), *n* = 8 (female *Bcl2l1*^fl/flCre^). * *p* < 0.05, ** *p* < 0.01, *** *p* < 0.001.

**Figure 3 ijms-24-17319-f003:**
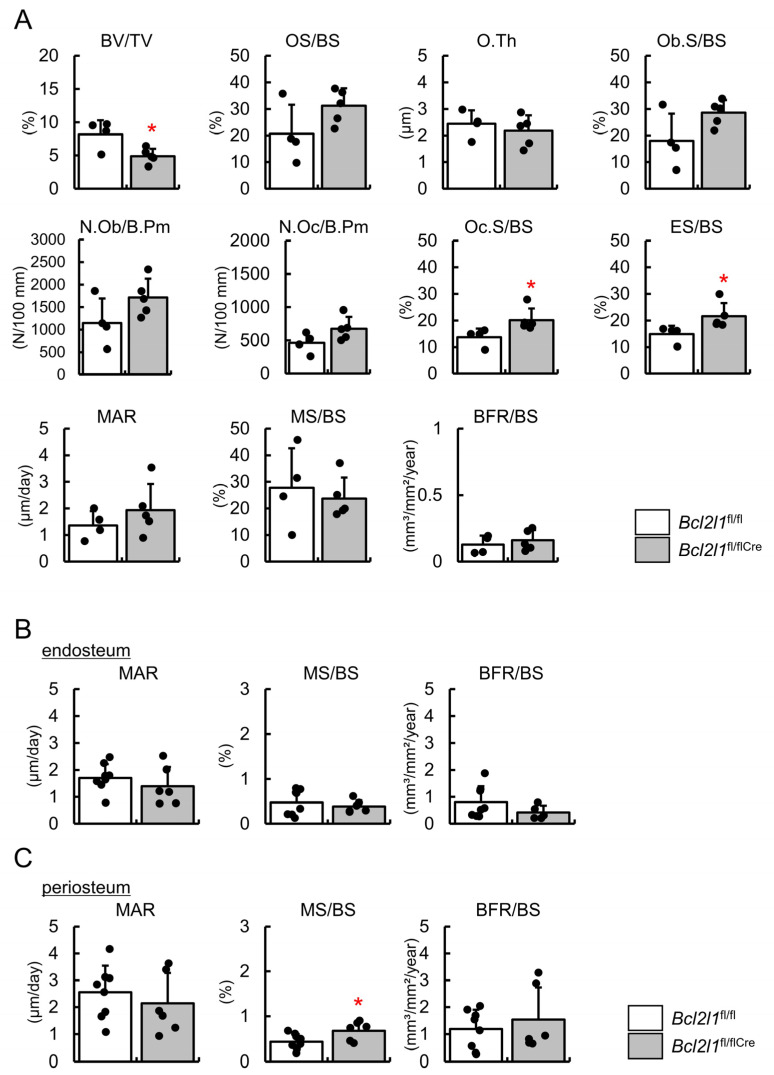
Bone histomorphometric analysis of femurs in male *Bcl2l1*^fl/fl^ and *Bcl2l1*^fl/flCre^ mice at 10 weeks of age. (**A**) Bone histomorphometric analysis of trabecular bone. Trabecular bone volume (bone volume/tissue volume, BV/TV), osteoid surface (OS/BS), osteoid thickness (O.Th), osteoblast surface (Ob.S/BS), number of osteoblasts (N.Ob/B.Pm), number of osteoclasts (N.Oc/B.Pm), osteoclast surface (Oc.S/BS), eroded surface (ES/BS), mineral apposition rate (MAR), mineralizing surface (MS/BS), and bone formation rate (BFR/BS) were compared. (**B**,**C**) Dynamic bone histomorphometric analyses of the endosteum (**B**) and periosteum (**C**) of the cortical bone. *n* = 8 (*Bcl2l1*^fl/fl^), *n* = 6 (*Bcl2l1*^fl/flCre^). * *p* < 0.05.

**Figure 4 ijms-24-17319-f004:**
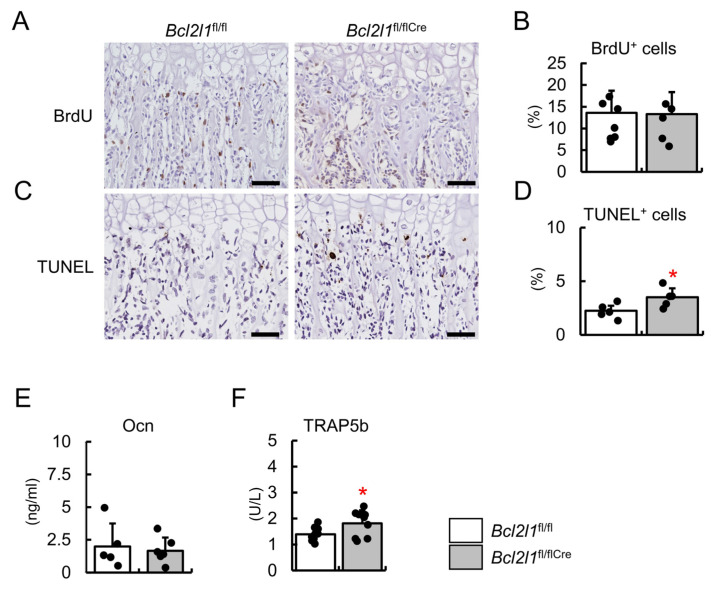
Osteoblast proliferation and apoptosis and serum markers for bone formation and resorption. (**A**–**D**) BrdU labeling (**A**) and TUNEL staining (**C**) of sections of the distal femoral metaphysis from *Bcl2l1*^fl/fl^ and *Bcl2l1*^fl/flCre^ mice at 2 weeks of age. Sections were counterstained with hematoxylin. BrdU-positive osteoblastic cells and TUNEL-positive cells were counted and shown as a percentage of the number of osteoblastic cells (**B**,**D**). *n* = 5–7. Scale bars: 50 μm (**A**,**C**). (**E**,**F**) Serum osteocalcin (Ocn) (**E**) and TRAP5b (**F**) in male *Bcl2l1*^fl/fl^ and *Bcl2l1*^fl/flCre^ mice at 10 weeks of age. Ocn: *n* = 5 (*Bcl2l1*^fl/fl^), *n* = 6 (*Bcl2l1*^fl/flCre^), TRAP5b: *n* = 9 (*Bcl2l1*^fl/fl^), *n* = 9 (*Bcl2l1*^fl/flCre^). * *p* < 0.05.

**Figure 5 ijms-24-17319-f005:**
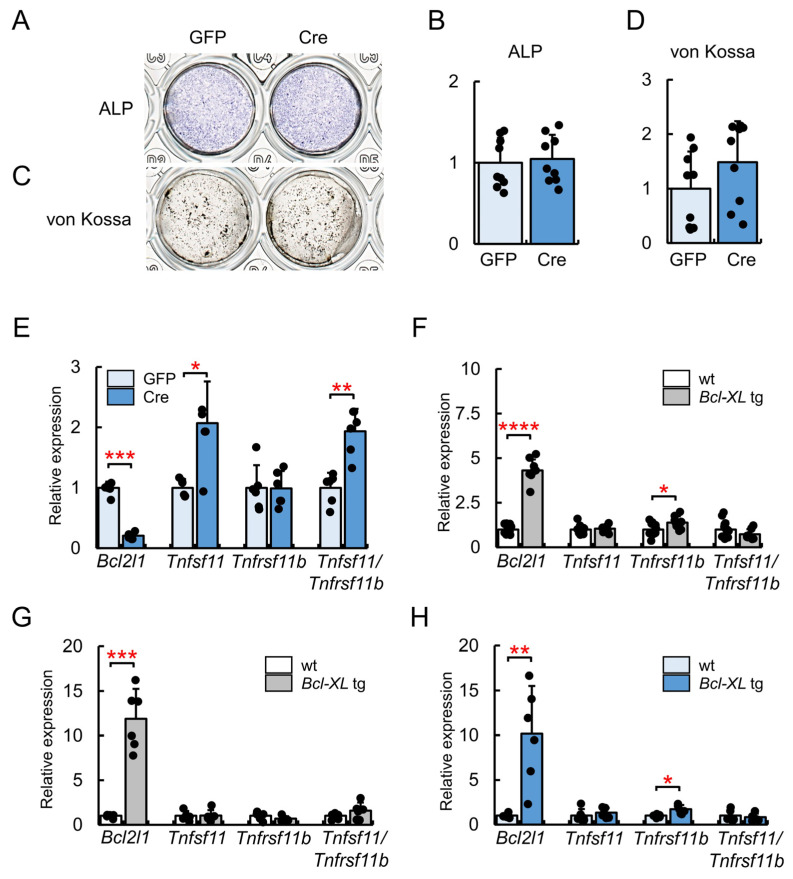
In vitro osteoblastogenesis. ALP (**A**) and von Kossa (**C**) staining, their quantification (**B**,**D**), and a real-time RT-PCR analysis (**E**). Primary osteoblasts from *Bcl2l1*^fl/fl^ mice were infected with a GFP- or Cre-expressing adenovirus. ALP and von Kossa staining was performed after the culture for 3 and 12 days, respectively, in the osteogenic medium, and RNA was extracted after the culture for 4 days in the osteogenic medium. Values for the GFP-expressing adenovirus were defined as 1 and relative levels are shown. *n* = 9 in (**B**,**D**), *n* = 6 in (**E**). Similar results were obtained from two (**A**–**D**) and four (**E**) independent experiments and representative data are shown. (**F**,**G**) Real-time RT-PCR analysis of *Tnfsf11* and *Tnfrsf11b* expression in *Bcl-XL* transgenic (tg) mice. RNA was prepared from the newborn calvariae of wild-type (*n* = 15) and *Bcl-XL* tg (*n* = 9) mice (**F**), and from osteoblast-enriched samples from the femurs and tibiae of wild-type (*n* = 5) and *Bcl-XL* tg (*n* = 6) mice at 6 weeks of age (**G**). (**H**) Real-time RT-PCR analysis of *Tnfsf11* and *Tnfrsf11b* expression in primary osteoblasts from wild-type (*n* = 6) and *Bcl-XL* tg (*n* = 6) mice. * *p* < 0.05, ** *p* < 0.01, *** *p* < 0.001, **** *p* < 0.0001.

**Figure 6 ijms-24-17319-f006:**
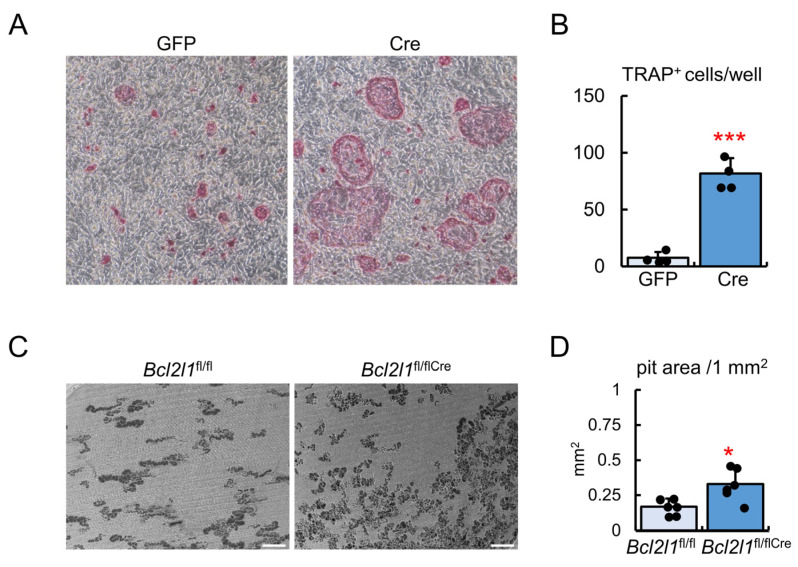
In vitro osteoclastogenesis. (**A**,**B**) TRAP staining (**A**) and the number of TRAP-positive cells (**B**). Primary osteoblasts from *Bcl2l1*^fl/fl^ mice were infected with the GFP- or Cre-expressing adenovirus, and infected cells were co-cultured with BMMs from *Bcl2l1*^fl/fl^ mice for 4 days. The number of multinucleated TRAP-positive cells in 48-well plates is shown in (**B**). *n* = 4. (**C**,**D**) Pit assay. Primary osteoblasts from *Bcl2l1*^fl/fl^ or *Bcl2l1*^fl/flCre^ mice were co-cultured with BMMs from *Bcl2l1*^fl/fl^ mice on dentin slices for 8 days (**C**). The pit area in 1 mm^2^ was measured (**D**). *n* = 6. Scale bars: 200 μm (**C**). Similar results were obtained from two independent experiments and representative data are shown. * *p* < 0.05, *** *p* < 0.001.

**Figure 7 ijms-24-17319-f007:**
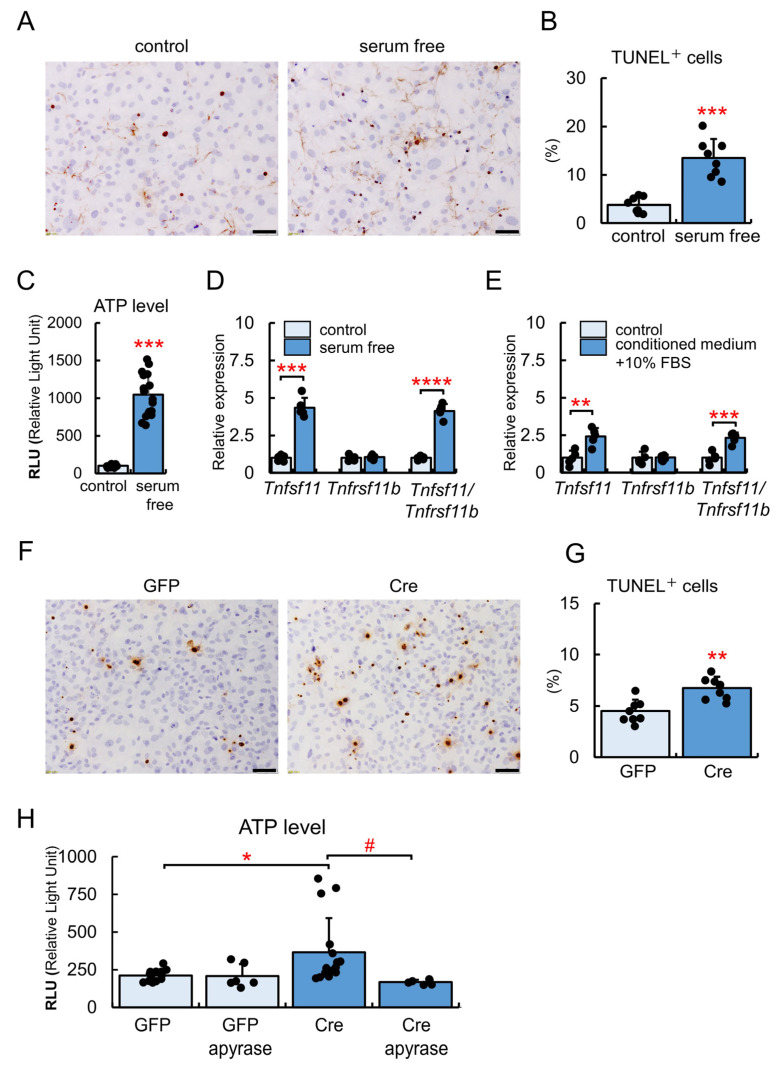
TUNEL-staining and measurement of ATP in vitro. (**A**–**C**) Primary osteoblasts from wild-type mice were cultured in the presence or absence of 10% FBS for 24 h and then stained for TUNEL (**A**). The frequencies of TUNEL-positive cells were calculated (**B**). The amount of ATP in the culture medium was quantitated by luciferase (**C**). *n* = 8. (**D**,**E**) Real-time RT-PCR analysis of *Tnfsf11* and *Tnfrsf11b*. Primary osteoblasts were cultured in the presence or absence of 10% FBS for 24 h (**D**). The supernatant with 10% FBS was used for the culture of primary osteoblasts (control), and the supernatant without 10% FBS was used as conditioned medium in the culture of primary osteoblasts after the addition of FBS to 10% (conditioned medium + 10% FBS) (**E**). *n* = 5. (**F**–**H**) Primary osteoblasts from *Bcl2l1*^fl/fl^ mice were infected with the GFP- or Cre-expressing adenovirus. After 48 h, cells were stained for TUNEL (**F**), and the frequencies of TUNEL-positive cells were calculated (**G**). The amount of ATP in the culture medium with or without apyrase was quantitated (**H**). *n* = 13 (GFP), *n* = 6 (GFP + apyrase), *n* = 16 (Cre), *n* = 6 (Cre + apyrase). *, ^#^
*p* < 0.05, ** *p* < 0.01, *** *p* < 0.001, **** *p* < 0.0001. Scale bars: 50 μm. Similar results were obtained from two independent experiments and representative data are shown.3. Discussion.

## Data Availability

Data are contained within the article.

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
