# Peer review of "Bcl2l1 Deficiency in Osteoblasts Reduces the Trabecular Bone Due to Enhanced Osteoclastogenesis Likely through Osteoblast Apoptosis"

_ijms, 2023, doi:10.3390/ijms242417319_

Round 1

Reviewer 1 Report

Comments and Suggestions for Authors

In this manuscript, Takeshi Moriishi and colleagues aim to elucidate the functions of Bcl2l1 in osteoblasts and bone homeostasis They discovered that Bcl2l1 was involved in the survival of osteoblasts, the Bcl2l1 deletion in osteoblasts appeared to enhance bone resorption by inducing osteoblast apoptosis, and ATP released from apoptotic osteoblasts may be responsible for enhanced bone resorption. Overall, this is a highly intriguing study, although there are a few small issues that require attention.

1.      Abstract: the reason why Bcl2l1 was selected to study need to clarify.

2.      Figure 1D, the qualification of WB is needed.

3.      The typing errors need to be corrected. Such as line 228, line 233, line 240.

4.      the quality of the pictures needs a higher one. Such as Figure 1B, Figure 5A

Comments on the Quality of English Language

Minor editing of English language required

Reviewer 2 Report

Comments and Suggestions for Authors

The authors herein demonstrated that osteoblast numbers and bone formation were maintained in osteoblast-specific Bcl2l1-deficient mice, whereas osteoclastogenesis was accelerated through the induction of Tnfsf11expression in osteoblasts likely through the release  of ATP from apoptotic osteoblasts, resulting in a reduction in trabecular bone.

The introduction describes the problem taking into account the different options. The cited references are adequate

The results are clearly described following a pathophysiological pattern. The discussion is adapted to the results obtained. Authors should indicate the strengths and limitations of the study The results are clearly described allowing them to be repeated by another research group. The statistical methodology is poorly described

Round 2

Reviewer 2 Report

Comments and Suggestions for Authors

The authors have answered the questions